# Low Threshold Plasmonic Nanolaser Based on Graphene

**Litu Xu, Fang Li \*, Shuai Liu, Fuqiang Yao and Yahui Liu**

Optical Information Technology Lab, School of Optoelectronics and Energy, Wuhan Institute of Technology, Wuhan 430205, China; 15871430579@163.com (L.X.); 15342251052@163.com (S.L.); yfq15872268171@163.com (F.Y.); 15872431358@163.com (Y.L.)
\* Correspondence: lifang@wit.edu.cn or lifang_wit@hotmail.com; Tel.: +86-027-8799-2024



**Featured Application: The paper describes the theoretical modelling of a surface plasmon laser, based on Ag and CdS nanowires spaced with either SiO₂ and graphene. A quite extensive parametric study is carried out, exploring the effects of the variation of the geometrical parameters. Results of the study are compared with the references. The most relevant result of the investigation is that the use of graphene as a space layer has apparently a very positive impact on the losses of the structure, and to lower the threshold for plasmon oscillation.**

**Abstract:** A hybrid plasmonic nanolaser based on nanowire/air slot/semicircular graphene and metal wire structure was designed. In this structure, the waveguides in the nanowires and the graphene-metal interface are coupled to form a hybrid plasma mode, which effectively reduces the metal loss. The mode and strong coupling of the laser are analyzed by using the finite-element method. Its electric field distribution, propagation loss, normalized mode area, quality factor, and lasing threshold are studied with the different geometric model. Simulation results reveal that the performance of the laser using this structure can be optimized by adjusting the model parameters. Under the optimal parameters, the effective propagation loss is only 0.0096, and the lasing threshold can be as low as $0.14~\mu m^{-1}$. This structure can achieve deep sub-wavelength confinement and low-loss transmission, and provides technical support for the miniaturization and integration of nano-devices.

**Keywords:** hybrid plasma; nanolaser; finite element; threshold; mode properties

## 1. Introduction

Since the advent of the first laser in the 1960s, lasers, like other major human inventions, have had a huge impact on human production and life. In recent years, nanotechnology has developed rapidly and has had a profound impact on information and materials. Photonic devices, including lasers, have also become more miniaturized and highly integrated [1,2]. However, subject to the diffraction limit, in the traditional semiconductor laser structure, its spatial size and mode size are larger than half a wavelength, and the miniaturization of the laser is hindered, which seriously restricts its integration with nano-optical devices [3–9]. Nano-lasers based on surface plasmons (SPs [10–14]) are known as the world's smallest lasers because of their spatial size and mode size that simultaneously break through the diffraction limit [15]. In 2007, Hill et al. first designed and verified metal-coated semiconductor nanolasers corresponding to wavelength electro-injection [16,17]. In 2009, Rupert F. Oulton et al. achieved surface plasmon laser lasing by experiments, which created a new method for exploring the interaction between light and matter [18,19]. Although SPs have broad application prospects in many fields, there are still many problems to be solved in practical applications.

These SPs waveguides have large loss during transmission because their structure contains a metal material having a negative dielectric constant, resulting in a relatively short transmission distance of light. How to make the waveguide structure have a lower propagation loss and have a super light-field local limiting ability is an urgent problem to be solved in the field of nanophotonics. Hybrid plasmonic waveguide (HP-waveguide) structure combines the low-loss transmission characteristics of dielectric optical waveguides with the strong optical field limitation of SPs waveguides [20–26], which has caused a new upsurge in the research field. In addition, with the advent of new materials, such as perovskite, graphene, and $MoS_2$, the structural properties have been further improved. Representatively, in 2017, Haichao Yu and others of Harbin Institute of Technology used organic-inorganic hybrid perovskite as the gain medium nanowire material, which effectively reduced the cavity loss [27–32]. However, there is still a large material loss, which directly affects the further reduction of the laser threshold.

In this paper, a new material, graphene, with excellent optical properties, and a low refractive index air slot are introduced to couple the gain dielectric waveguide and the surface plasmon wave of the metal with graphene to form a new hybrid plasma waveguide structure. In the nanolaser design, the metal loss is effectively reduced, and the threshold of the laser is lower. The finite element method is used to analyze the variation of the mode characteristics, quality factor, and threshold of the laser with the geometric model, and the model of the nanolaser is optimized to make the overall performance better.

## 2. Theoretical Analysis

SPs is an electromagnetic energy wave formed on the metal surface when the external light field is incident on the interface of the metal medium, and the free electrons in the metal are redistributed under the action of electromagnetic waves and collective oscillation occurs. Theoretical analysis shows that the SPs wave only has the TM polarization mode on the metal-medium surface, and its dispersion relation is as follows [33]:

$$k_{sp} = \frac{\omega}{c} \sqrt{\frac{\varepsilon_m \varepsilon_d}{\varepsilon_m + \varepsilon_d}} \tag{1}$$

In the above formula, $\varepsilon_m$ and $\varepsilon_d$ are the dielectric constants of the metal and the medium, respectively, and $\omega/c$ is the wave vector of the light in the air. The wave vector of the incident light field is always smaller than the propagating wave vector of SPs, which is the basic property of the SPs wave. By the phase shift method [34], the surface plasmons can be effectively excited only when the two vectors are matched.

Due to the ohmic loss in the metal, the energy in the surface wave will be attenuated as the propagation distance increases. The propagation length of the SPs wave is calculated as:

$$L_{spp} = \lambda_0 \frac{(\varepsilon'_m)^2}{2\pi\varepsilon''} \left( \frac{\varepsilon'_m + \varepsilon_d}{\varepsilon'_m \varepsilon_d} \right)^{3/2} \tag{2}$$

where $\varepsilon_m'$ is the real part of the dielectric constant of the metal, and $\varepsilon_m''$ is the imaginary part of the dielectric constant of the metal. It can be seen from the above formula that in order to obtain a large transmission length, the real part of the dielectric constant of the metal material should be larger, and the imaginary part should be smaller. Precious metal has a low loss and a low absorption coefficient and satisfies the above conditions; therefore, precious metals are generally selected. According to the references, gold and silver are generally used [35,36], and silver is used as a metal in the design.

## 3. Physical Model

The structure of the hybrid plasma laser designed in this paper is shown in Figure 1. The nanolaser consists of a gain medium nanowire, left $SiO_2$ layer, right $SiO_2$ layer, metal nanowire, graphene nanoribbon, and air gap between the gain medium and graphene.

First, the graphene was transferred to the SiO$_2$ layer by the computer stripping method to form a 5 nm thick graphene nanobelt, and then the Ag was evaporated under graphene to form semicircular metal nanowires. Using focused ion beam technology (FIB), an air gap with a width of $w_g$ and a thickness of $h_t$ on the surface of the SiO$_2$ layer was etched, and $h_t$ is the distance from the bottom end of the dielectric nanowire to the top of the graphene. Finally, a CdS nanowire as a gain medium is placed along the axial direction of the air slot by a micro-nano operation. In order to enlarge the coupling area between the gain dielectric waveguide and the SP wave, the ends of the nanowire constituting the F-P cavity are circular. The length, L, of the CdS nanowire is 10 μm, the radius is fixed at 80 nm, and the radius of the semicircular Ag is 90 nm. Most related studies have analyzed the influence of the radius of the nanowire, so this article is not discussed. The width and height of the air slots in the design can vary over a certain range.

In order to reduce the absorption and heat loss of the metal, the graphene layer and the air gap are used to separate the CdS nanowires from the metal, Ag. This is because the graphene nanoribbons have good thermal conductivity as a buffer layer and can store localized electric field energy caused by dielectric waveguides and SP coupling. In addition, Ag is easily oxidized and corroded in the air. After being wrapped with graphene nanoribbons, it can protect the laser, effectively reduce the transmission loss of the laser, and improve its working stability.

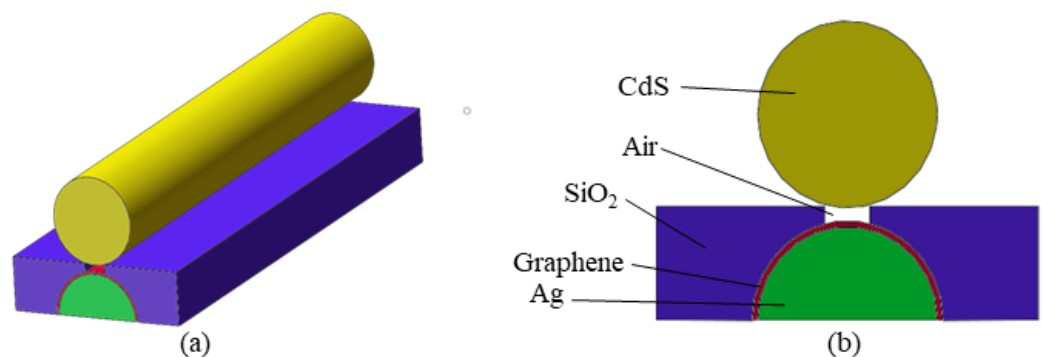

**Figure 1.** (**a**) The block diagrams of the nanolaser. (**b**) The section plan of the nanolaser.

## 4. Numerical Simulation and Analysis

The hybrid plasma laser has an output wavelength of 490 nm [37], and the dielectric constants of the CdS and SiO$_2$ materials are 5.76 and 1.96, respectively. The dielectric constants of graphene and silver are analyzed by the Drude model, and the calculated values are 2.2 + 4.8i and −9.2 + 0.3i [38,39]. The finite element method is used to establish the mathematical model. By using the continuous boundary condition internally, using the scattering boundary condition in outer boundary, refining the mesh at the junction of the air slot and the nanowire, and improving the mesh quality, the calculation accuracy is ensured [40]. Thus, the mode characteristics and gain thresholds in the structure are calculated.

### 4.1. Discussion of Electric Field Distribution

The normalized electric field distribution of the two-dimensional and three-dimensional models of the laser was simulated by the finite element method. The simulation results are shown in Figures 2 and 3. It can be seen from Figure 2a that the field enhancement effect at the air groove between the nanowire and the graphene is remarkable, and the high localization of the energy constraint is achieved. This is because the continuous movement of electrons on the surface of the metal film causes a strong light field to be generated there, and at the air gap between the metal Ag and the nanowire, the semiconductor nanowire waveguide mode is coupled with the surface plasmon mode of the metal interface, resulting in a localized effect of partial energy. Figure 2b is a normalized electric field distribution of the horizontal serif in Figure 2a. It can be seen from Figure 3 that under

the excitation of the pump light, which is lased by a titanium sapphire laser with a center wavelength of 405 nm, a repetition rate of 10 kHz, and a pulse width of 100 fs [41], the SP wave generated on the metal surface is coupled into the CdS nanowire through the graphene and the air gap. The CdS nanowire is used as the gain medium and also the FP resonator, so the partial SP waveguide and the high gain dielectric waveguide are coupled and oscillated from both ends of the nanowire after oscillation amplification in the cavity.

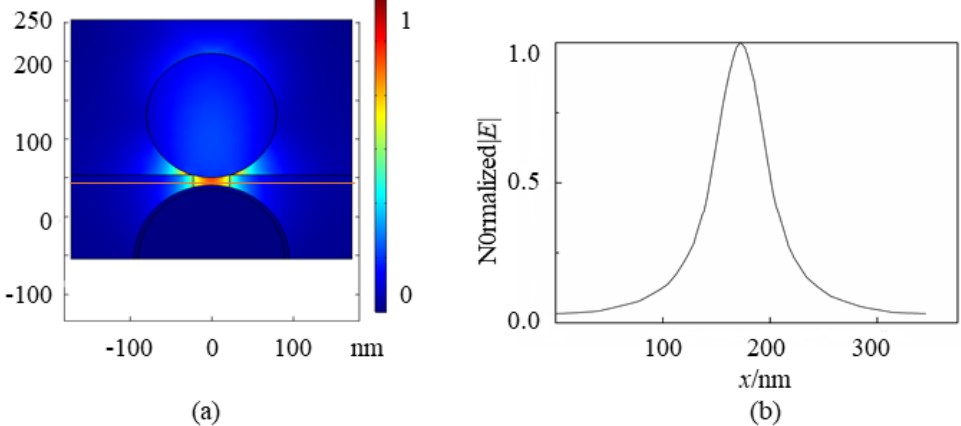

**Figure 2.** (**a**) 2D electric field distribution of the nanolaser. (**b**) Horizontal serif electric field distribution.

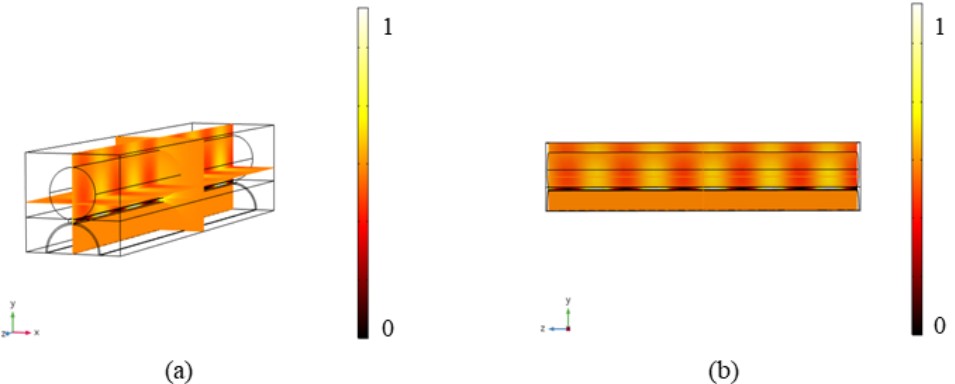

**Figure 3.** (**a**) Three-dimensional mode of the nanolaser and the (**b**) y-z normalized electric field.

*4.2. Model Characteristics Analysis*

In order to optimize the structure designed in this paper and apply it to the nanolaser, its mode characteristics, gain threshold, and quality factor (Q) were analyzed. The main parameters reflecting the mode characteristics are the effective refraction index ($n_{\text{eff}}$), the effective propagation loss ($\alpha_{\text{eff}}$), the normalized mode area (*SF*), and the confinement factor ($\Gamma$). Among them, the imaginary part of the effective index characterizes the propagation loss of the mode. The normalized mode area represents the constraint ability of the mode field. The calculation formula is [41,42]:

$$SF = \frac{A_{\text{eff}}}{A_0} = \frac{\left(\iint |E|^2 dx dy\right)^2 / \left(\iint |E|^4 dx dy\right)}{\lambda^2/4} \tag{3}$$

where $A_{\text{eff}}$ is the effective mode area, $A_0$ is the mode area under diffraction limit, and $\lambda$ is the wavelength in vacuum. The confinement factor, $\Gamma$, is defined as the ratio of the electric field energy in

the CdS nanowire to the electric field energy of the entire mode, which is used to characterize the field strength limiting ability of the gain medium nanowire. The calculation formula is:

$$\Gamma = \frac{W_s}{W} \tag{4}$$

where $W_s$ represents the energy stored in the gain medium, and $W$ represents the total energy of the mode field.

Figure 4a–d show the variation of the mode characteristics of the nanolaser with the air gap width, $w_g$, and the distance, $h_t$, from the bottom of the nanowire to the top of the graphene. As can be seen from Figure 4a,b, when the width of the air gap increases, the effective refraction index decreases gradually; the propagation loss of the mode decreases first and then increases. At $w_g = 35$ nm, the minimum value is 0.0096, and the overall average value is 0.013. This is because when the width of the air slot is small, although the distance from the bottom of the nanowire to the top of the graphene remains unchanged, the overall relative height of the two is large, and the SP mode is mainly concentrated on the metal surface, causing a large loss. As the width of the air gap increases, the relative height decreases, and the SP mode coupling effect of the gain dielectric waveguide is enhanced, and the loss is gradually reduced. When the air gap reaches an optimum value, the coupling effect is no longer significant, and the continued expansion of the air gap causes an increase in loss. As can be seen from Figure 4c,d, the increase in the air gap causes the normalized mode area to gradually decrease and the confinement factor to gradually increase. It can be seen that the normalized mode area is much smaller than 0.1, the minimum value is 0.0032, and the average value is 0.018, which indicates that the hybrid plasma laser structure has a strong ability to constrain the light field and can achieve a deep sub-wavelength constraint; the confinement factor is greater than 0.55, indicating that the mode field energy is mainly located in the gain medium nanowire, and the hybrid plasma waveguide propagates in the nanowire.

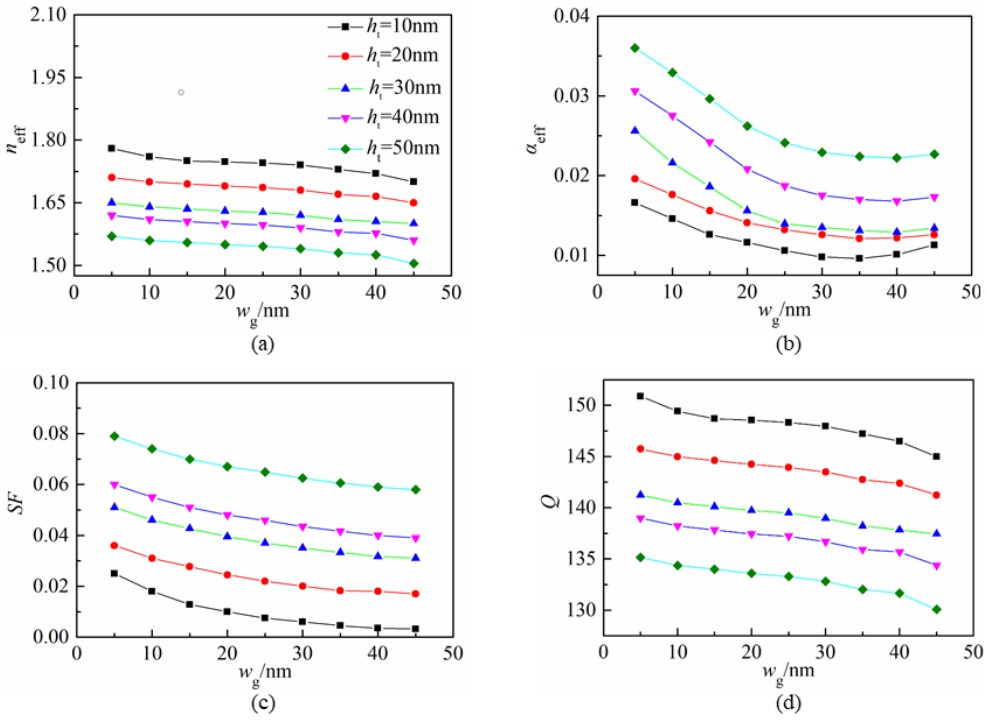

**Figure 4.** The mode properties change with the air gap, $w_g$, and the distance, $h_t$; (**a**) effective index, $n_{eff}$; (**b**) effective propagation loss, $\alpha_{eff}$; (**c**) normalized mode area, SF; (**d**) confinement factor, $\Gamma$.

At the same time, it can be seen that the refraction effective index and the confinement factor gradually decrease, as the distance, $h_t$, from the bottom end of the nanowire to the top of the graphene increases, while the propagation loss and the normalized mode area increase.

### 4.3. Analysis of Q-Value and Gain Threshold

The quality factor, $Q$, is an important parameter for evaluating the performance of the cavity, reflecting the ability of the cavity to bind to the photon. The larger the $Q$ value is, the longer it stores photons, and its expression is [43]:

$$Q = 2\pi f \tau_R = 2\pi f \frac{L}{\delta c} \tag{5}$$

where $f$ is the frequency of the optical field in the cavity, $\tau_R$ is the time constant of the cavity, $\delta$ is the loss in the cavity, $L$ is the length of the cavity, and c is the speed of light in the vacuum.

The threshold ($g_{th}$) is the minimum gain required to satisfy the laser's realization of stimulated radiation and is an important basis for reflecting the performance of the laser. The lower the threshold is, the less gain the laser needs to achieve lasing, and the higher the quality of its work is. The $g_{th}$ expression is:

$$g_{th} = [k_0 \alpha_{eff} + \ln(1/R)/L]/\Gamma(n_{eff}/n_{wire}) \tag{6}$$

where $k_0 = 2\pi/\lambda$ is the wave number in vacuum, $n_{wire}$ is the refractive index of the CdS nanowire, $n_{eff}$ is the real part of the effective index of the mode, $\Gamma$ is the confinement factor, and the scaling factor, $n_{eff}/n_{wire}$, is the enhancement part of the effective index of the mode. $R$ is the end face reflectivity, defined as [44,45]:

$$R = (n_{eff} - 1)/(n_{eff} + 1) \tag{7}$$

As can be seen from Figure 5a, the gain threshold, $g_{th}$, decreases first and then increases with the expansion of the air gap width. $g_{th}$ gradually increases with the increase of $h_t$ when $w_g$ is constant. At $w_g$ = 35 nm, $h_t$ = At 5 nm, a minimum value of 0.14 $\mu m^{-1}$ is obtained because the escalation in the depth of the air gap increases the loss of plasmon on the metal surface, which requires more optical gain, so the threshold is increased.

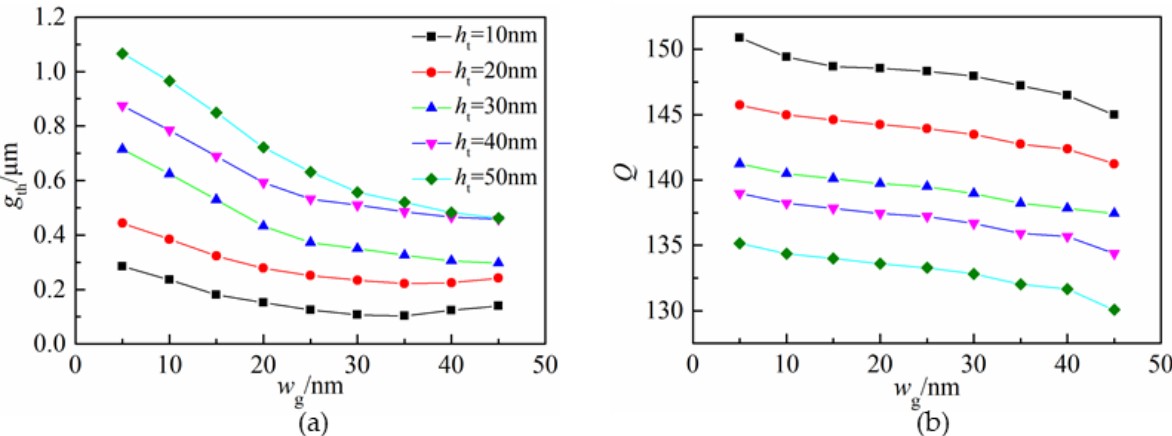

**Figure 5.** The change of gain threshold and quality factor with the structures (**a**); threshold, $g_{th}$; and (**b**) quality factor, $Q$.

Figure 5b shows that as the width of the air gap, $w_g$, increases, when the distance, $h_t$, from the bottom of the fixed nanowire to the top of the graphene is constant, the quality factor, $Q$, decreases gradually. When the $w_g$ is constant, the quality factor decreases as the $h_t$ increases. In the selected range of values, the maximum value is 150.9, indicating that the resonant cavity has a stronger ability to limit photons.

In summary, in order to optimize the performance of the nanolaser, the influence of geometric parameters on its mode characteristics, quality factor, and gain threshold are considered as a whole. Finally, the air gap width, $w_g$, is chosen to be 35 nm, and the distance, $h_t$, from the bottom of the nanowire to the top of the graphene is 10 nm. At this time, the laser has a propagation loss of 0.0096, an effective mode area of $0.001\lambda^2$, a threshold as low as 0.14 $\mu m^{-1}$, and a quality factor of 147.21, so that a low loss, low threshold, and deep subwavelength-constrained lasing can be achieved.

## 5. Conclusions

In this paper, a hybrid plasma nanolaser structure based on gain medium nanowires, air slot, semicircular graphene, and metal wires was proposed. Based on the finite element method, the simulation design was carried out. The effects of the air slot width and air gap depth on the mode characteristics, gain threshold, and quality factor were studied. The simulation results show that when the width of the air slot is 35 nm and the distance from the bottom of the nanowire to the top of graphene is 5 nm, the laser has the best performance. The propagation loss is 0.0096, the normalized mode area is 0.004, the threshold value is 0.14 $\mu m^{-1}$, the quality factor is 147.21, and the limiting factor is above 0.5. The low threshold indicates that the laser needs less gain to achieve lasing, and has a higher work quality. The nanolaser designed in this paper has a smaller size, better comprehensive performance, and can achieve ultra-deep sub-wavelength constraints and low loss transmission. It provides theoretical and technical support for the optimal design of plasma nanolaser, and has broad application prospects in biology, medicine, and other fields.

**Author Contributions:** L.X. and Y.L. helped proceeding the simulation processes and data analysis; F.L. organized the paper and encouraged in paper writing; Also, F.Y. and S.L. helped proceeding the simulation processes.

**Funding:** This research was funded by the National Natural Science Foundation of China (Grant No. 11204222), the Natural Science Foundation of Hubei Province, China (Grant No. 2013CFB316, Grant No. 2014CFB793), the Innovation Fund of School of Science, Wuhan Institute of Technology (No. CX2016106).

**Acknowledgments:** L.X. would like to thank the support of the Optical Information Technology Lab, Wuhan Institute of Technology.

**Conflicts of Interest:** The authors declare no conflict of interest.

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
