# Peer review of "Low Threshold Plasmonic Nanolaser Based on Graphene"

_applsci, doi:10.3390/app8112186_

Reviewer 1 Report

In this contribution, Xu et al. designed a hybrid plasmonic nanolaser and studied their properties using mathematical simulation. The authors demonstrated that the laser output can be optimized by adjusting the width of the air slot and the thickness of graphene on the surface of the silver nanowire. This work offers new insights into the design of plasma nanolaser. The presentation of the obtained results is clear. In my opinion, it is an interesting work which should be published. Before possible publication it would be better if the authors address the following minor comments;

1.      Line 22 in abstract and line 199 in conclusion, 0.14µm-1 should be expressed properly.

2.    Error bars are required for Figures 4 and 5.

3.      In page 4 line 16, the authors said that the pump laser was used to excite the nanolaser but they haven’t specified what wavelength and power they used. 

Author Response

1. Line 22 in abstract and line 199 in conclusion, 0.14µm-1 should be expressed properly.

Answers: We have made a specific statement in line 200-201, and it has been marked in yellow.

    2. Error bars are required for Figures 4 and 5.

Answers: Thanks to the reviewer for comments. This paper uses commercial COMSOL Multiphysics software based on the finite element method for the simulation calculation. Mode analysis calculation is under the frequency domain module of electromagnetic wave. Under certain parameters, its calculation results are stable and the error is negligible. Although the data selection has error precision, there is no obvious undulation forming error bars.

3. In page 4 line 16, the authors said that the pump laser was used to excite the nanolaser but they haven’t specified what wavelength and power they used.

Answers: We have made a specific statement in line 116-118. This paper mainly carries out simulation. For the type of pump light, we cited the reference [41] in our revised manuscript. In the reference [41], Zhang xiang's research group proposed a nano-laser with CdS-medium-Ag structure. The laser device was pumped using a titanium sapphire laser with a center wavelength of 405 nm, a repetition rate of 10 kHz, and a pulse width of 100 fs.

Reviewer 2 Report

This manuscript presents numerical calculation (using finite-element method) of the electrical field inside some nano-structure which they call as “nano-laser based on graphene”.  As I understand, the specific characteristics of laser like gain threshold, quality factor, normalized mode area, etc the authors have obtained from the electric field distribution with using some standard formulae from general optics. I can recommend this manuscript for publication but only in revised form.
Comments:
1) It is not clear why this nanostructure which gives some-where high level of the electric field the author call as nano-laser. It is really a laser. In their pioneering  work Bergman and Stockman have called it as SPASER in order to emphasize that it is not exactly a laser though a little similar. There are also several other articles of Bergman about SPASER which should be mentioned.
2) The formula (1) is not derived in Ref [33] but much and much earlier. The author should give the proper reference.
3) The sentence in the bottom of page 2 (lines 76-77) is not clear—it should be corrected.
4) Page 3, line 84: the word Using should be written from capital letter.
5) Caption to Fig. 2 (b). Is the word “serif” a physical scientific terminology?
6) In the end of Eq. (1) it should be point. In the end of Eq. (2) it should be comma and then from the new line it should be to written the word “where”. The same in Eq. (3), (4) etc.
7) In page 6 in line 170 the sub-index 0 in k0 should be written correctly.
8) The authors should better explain what the laser optics characteristics mean. In particular they did not explain what the confinement factor Gamma (see page 4, line 129) is.
9) What is effective index n_eff (see page 4, line 128)? Is it effective refraction index?
10) It is not explained how the authors take into account the graphene layer. Why it important. What is it thickness (graphene is strictly 2D) compare to size of the unit cell of the grid using in calculation? What will happen if do not use the graphene layer in the nanostructure?

Author Response

1. It is not clear why this nanostructure which gives some-where high level of the electric field the author call as nano-laser. It is really a laser. In their pioneering work Bergman and Stockman have called it as SPASER in order to emphasize that it is not exactly a laser though a little similar. There are also several other articles of Bergman about SPASER which should be mentioned.

Answers: Thank you very much for your comments. We have already explained high level of the electric field in somewhere (line115-118). SPASER(Surface Plasmon Amplification by Stimulated Emission of Radiation) or plasmonic laser is a type of laser which aims to confine light at a subwavelength scale far below Rayleigh's diffraction limit of light, by storing some of the light energy in electron oscillations called surface plasmon polaritons. This paper proposed a plasmonic nanolaser. We have made changes to the title of the article.

2. The formula (1) is not derived in Ref [33] but much and much earlier. The author should give the proper reference.

Answers: We have made changes to the references. This formula is in reference “RAETHER H. Surface Plasmons on Smooth and Rough Surfaces and on Gratings. Springer Berlin Heidelberg1988, 6-8. [Google Scholar]”.

3. The sentence in the bottom of page 2 (lines 76-77) is not clear—it should be corrected.

Answers: We have already modified this sentence ,and it has been marked in yellow.line75-78

4. Page 3, line 84: the word Using should be written from capital letter.

Answers: We have modified this word.(line87)

5. Caption to Fig. 2 (b). Is the word “serif” a physical scientific terminology?

Answers: No, it isn’t a physical scientific terminology. Its meaning is transversal line. Fig. 2(b) is a normalized electric field distribution of the horizontal serif in Fig. 2(a). There is a yellow transversal line in Fig.2(a).

6. In the end of Eq. (1) it should be point. In the end of Eq. (2) it should be comma and then from the new line it should be to written the word “where”. The same in Eq. (3), (4) etc.

Answers: We have revised the formula according to the above opinion, and it has been marked in yellow.

7. In page 6 in line 170 the sub-index 0 in k0 should be written correctly.

Answers: It has been modified to k0.

8. The authors should better explain what the laser optics characteristics mean. In particular they did not explain what the confinement factor Gamma (see page 4, line 129) is.

Answers: This article mainly studies electric field distribution, propagation loss, normalized mode area, quality factor and lasing threshold of the plasmonic laser. Their definitions are all given in this paper. Because of the length of the article, the definition of many terms in the article may not be very detailed. We have given the formula of confinement factor Gamma and explained it.(line144-145)

9. What is effective index n_eff (see page 4, line 128)? Is it effective refraction index?

Answers: Yes, it is effective refraction index. We have also made changes to other parts of the article.

10. It is not explained how the authors take into account the graphene layer. Why it important. What is it thickness (graphene is strictly 2D) compare to size of the unit cell of the grid using in calculation? What will happen if do not use the graphene layer in the nanostructure?

Answers: 1)In the line 96-101, there are explanations for why graphene is used. In the design, the new material graphene is used as the buffer layer, and a new hybrid plasma structure is formed with the metal and the gain medium, which effectively reduces the metal loss. In addition, the semi-circular graphene nanoribbon was used to wrap the silver nanowires, which solved the problem of easy oxidation and corrosion of silver in the air, improved its stability and reduced the size of the laser to a certain extent. 2) The thickness of single-layer graphene is about 0.35nm. It can’t be used in calculation compared to size of the unit cell of the grid. In this paper, multilayer graphene is used. Its thickness can reach about 5nm. 3) In order to improve the reliability and stability of the laser and reduce the metal loss, we choose graphene as buffer layer. In reference [39], there is a specific description of the photoelectric properties of graphene.
